# Mortality in Four Waves of COVID-19 Is Differently Associated with Healthcare Capacities Affected by Economic Disparities

**DOI:** 10.3390/tropicalmed7090241

**Published:** 2022-09-10

**Authors:** Lan Yao, J. Carolyn Graff, Lotfi Aleya, Jiamin Ma, Yanhong Cao, Wei Wei, Shuqiu Sun, Congyi Wang, Yan Jiao, Weikuan Gu, Gang Wang, Dianjun Sun

**Affiliations:** 1Department of Orthopedic Surgery and BME-Campbell Clinic, University of Tennessee Health Science Center, Memphis, TN 38163, USA; 2College of Nursing, University of Tennessee Health Science Center, Memphis, TN 38105, USA; 3Chrono-Environnement Laboratory, UMR CNRS 6249, Bourgogne Franche-Comté Université, CEDEX 21010, F-25030 Besançon, France; 4The First Affiliated Hospital of Harbin Medical University, 23 Youzheng Street, Nangang District, Harbin 150001, China; 5Center for Endemic Disease Control, Chinese Center for Disease Control and Prevention, Harbin Medical University, 157 Baojian Road, Harbin 150081, China; 6The Center for Biomedical Research, Department of Respiratory and Critical Care Medicine, NHC Key Laboratory of Respiratory Diseases, Tongji Hospital, Tongji Medical College, Huazhong University of Science and Technology, Wuhan 430030, China; 7Research Service, Memphis VA Medical Center, 1030 Jefferson Avenue, Memphis, TN 38104, USA

**Keywords:** COVID-19, economy, income levels, mortality, waves, policy, turning points

## Abstract

Background: The greatest challenges are imposed on the overall capacity of disease management when the cases reach the maximum in each wave of the pandemic. Methods: The cases and deaths for the four waves of COVID-19 in 119 countries and regions (CRs) were collected. We compared the mortality across CRs where populations experience different economic and healthcare disparities. Findings: Among 119 CRs, 117, 112, 111, and 55 have experienced 1, 2, 3, and 4 waves of COVID-19 disease, respectively. The average mortality rates at the disease turning point were 0.036, 0.019. 0.017, and 0.015 for the waves 1, 2, 3, and 4, respectively. Among 49 potential factors, income level, gross national income (GNI) per capita, and school enrollment are positively correlated with the mortality rates in the first wave, but negatively correlated with the rates of the rest of the waves. Their values for the first wave are 0.253, 0.346 and 0.385, respectively. The r value for waves 2, 3, and 4 are −0.310, −0.293, −0.234; −0.263, −0.284, −0.282; and −0.330, −0.394, −0.048, respectively. In high-income CRs, the mortality rates in waves 2 and 3 were 29% and 28% of that in wave 1; while in upper-middle-income CRs, the rates for waves 2 and 3 were 76% and 79% of that in wave 1. The rates in waves 2 and 3 for lower-middle-income countries were 88% and 89% of that in wave 1, and for low-income countries were 135% and 135%. Furthermore, comparison among the largest case numbers through all waves indicated that the mortalities in upper- and lower-middle-income countries is 65% more than that of the high-income countries. Interpretation: Conclusions from the first wave of the COVID-19 pandemic do not apply to the following waves. The clinical outcomes in developing countries become worse along with the expansion of the pandemic.

## 1. Introduction

A considerable amount of research has been conducted to better understand factors influencing mortality caused by the pandemic of COVID-19 [1,2,3,4,5,6,7]. Most data are based on the first wave, while some are based on the combination of the first and second waves of the pandemic [1,2,3,4,5,6,7]. None of these studies revealed the impact of income levels, with other factors, such as transportation and population density, regarded as the major predictive factors in the pandemic [6,7,8]. A report based on the second wave of the COVID-19 pandemic indicated that COVID-19 was more severe on the African continent than the first wave [1]. The question remains as to whether economic levels affect the mitigation of COVID-19 in stages such as in waves 2, 3, and 4.

Based on reports from Worldometers (https://www.worldometers.info/coronavirus/ (accessed on 14 January 2022)) [9] and the World Health Organization (WHO) Coronavirus (COVID-19) Dashboard (https://covid19.who.int/ (accessed on 14 January 2022)) [2], accessed in early 2022, more than 100 countries and regions (CRs) have experienced more than three waves of the disease. These data provide a chance to compare the influential factors in fighting COVID-19 in up to four waves of the disease among CRs at different economic levels.

In order to measure the overall capacity of CRs in fighting COVID-19, this study focuses on the time points of the mortality rate derived from two intrinsic turning points, the maximum of case and death numbers, during each wave of the pandemic. For each CR, we measure its capacity in the management of the disease by the mortality rate at the turning point of the disease.

Income level in a country or region reflects the level of resources and medical facilities that determine the overall capacity of fighting a pandemic. Because the COVID-19 disease became pandemic, the income level in a country or regional level was essential for fighting the disease. In particular, when the disease reaches a maximum level, it challenges the biomedical capability of a country or region. The mortality rate at the time of the disease peak days indicates whether the medical resource is good enough to deal with the hospitalized patients. This study examines the relationship between income level and mortality at the peak of the disease, and determines how different countries at different income levels fight COVID-19 pandemics.

## 2. Methods

### 2.1. Data Sources

We conducted correlation analyses between mortality rates at peak points in different waves and a total of 49 potential influential factors: comparisons of mortality rates at peak points among different levels of incomes, the mortality rates at the maximum case numbers among all waves in different income levels, and time- and case-adjusted mortality rates among different income levels. The average numbers of cases and deaths of seven days in the peak days of different waves of COVID-19 were collected from Worldometers, and confirmed with the WHO Coronavirus (COVID-19) Dashboard. Population statistics were collected from Worldometers (https://www.worldometers.info/world-population/population-by-country/ (accessed on 14 January 2022)). Data collection started on 21 January 2022 and ended on 2 February 2022.

### 2.2. Influential Factors

The information of potential factors from each CR was collected from the World Bank. A total of 49 potential influential factors were collected. These factors include→Forest area (% of land area), →Rural population (% of total population)→Surface area (sq. km), →Mortality rate, under-5 (per 1000 live births), →Incidence of tuberculosis (per 100,000 people), →Access to electricity (% of population), →CO_2_ emissions (metric tons per capita), →Urban population, →Energy use (kg of oil equivalent per capita), →Urban population (% of total population), →Adjusted net savings, including particulate emission damage (% of GNI), →Expense (% of GDP), →Foreign direct investment, net inflows (BoP, current USD), →Gross savings (% of GDP), →Exports of goods and services (% of GDP), →Imports of goods and services (% of GDP), →Government expenditure on education, total (% of GDP), →Literacy rate, youth total (% of people ages 15–24), →Unemployment, total (% of total labor force), →Population living in slums (% of urban population), →Cause of death, by communicable diseases and maternal, prenatal and nutrition conditions (% of total), →Refugee population by country or territory of origin, →Hospital beds (per 1000 people), →Prevalence of overweight, weight for height (% of children under 5), →Specialist surgical workforce (per 100,000 population), →Air transport, registered carrier departures worldwide, →Annual freshwater withdrawals, total (billion cubic meters), →Fixed telephone subscriptions (per 100 people), →Fixed broadband subscriptions (per 100 people), →Secure Internet servers (per 1 million people), International tourism, receipts (% of total exports),→Researchers in R&D (per million people), →PM2.5 air pollution, population exposed to levels exceeding WHO guideline value (% of total), →Population in the largest city (% of urban population), →Top countries in number of air passengers carried in 2019, →Ownership of passenger cars (units per thousand persons). Income levels of CRs were obtained from the World Bank’s Atlas method, which relies on the gross national income (GDP) per capita in 2019 at nominal values as an indicator of income.

### 2.3. Criteria of Data Collection

Inclusion criteria: (1) For the first three waves of the pandemic, data were collected from CRs with at least 100,000 reported total cases; (2) for wave 4, additional data were collected from CRs with a reported total case number of more than 50,000; and (3) at least one visible turning point was reported in at least one wave in both of the cases and deaths, judged by two authors. Exclusion criteria: (1) no obvious turning point in the data in either cases or deaths; (2) the number of deaths on the turning point was less than 3; and (3) the days of turning points between cases and death were more than 2 months. 

### 2.4. Definition of Waves and Turning Points

A disease wave is defined as the cases and deaths that had turning points (peak days) and were flanked on both sides by days with a smaller number of cases or deaths (Appendix A). The average number of seven days was used to define the turning point (or peak) of a wave [2]. The cases turning point was defined as the day with the largest number of average cases in seven days (the seven-day average). Case numbers on each side of the peak must decrease at least 10%, in comparison with the number on a peak day. The same criteria were also used to determine the peak of deaths. 

### 2.5. Definition of Mortality at the Day of the Turning Point

The mortality of each wave is defined as the average number of seven days at the death turning point divided by the average number of seven days at the case turning point in the same wave. When the number of days between a case turning point and a death turning point is the smallest, then these case and death points are considered to be in the same wave. If the days between these two turning points are more than two months, they are not regarded as the same wave, even when there is not any wave during the pandemic.

### 2.6. Data Uniformity and Bias Checking

Data collection was conducted by two investigators and double-checked by a third researcher. Outliers identified by individual authors were discussed by at least two authors. Peak days and data on peak days were double-checked by two additional authors. Statistical analyses were conducted by two authors to ensure accuracy. Wave numbers were adjusted based on timeline and case numbers to evaluate the accuracy of the data analysis. 

## 3. Statistical Analysis

Data analysis was conducted in the following ten steps. (1) CRs names, days of case peak in a wave, number of cases of a wave, days of death peaks, and number of deaths on the peak days were collected and stored in an Excel file. A total of 49 influence factors of CRs were collected in a separate Excel file. (2) Mortality at the peak of each wave was calculated by dividing the seven-day average number of the day of the death peak with the seven-day average number of cases on the day of the cases peak. (3) The correlation coefficient (r) between these mortality and influential parameters were calculated with the formula function of Excel. Student *t*-tests were conducted with paired comparison and two tailed distributions. (4) Influential factors with positive or negative impacts were selected for evaluation. (5) Mortalities in three waves in different income levels of CRs were analyzed and compared with each other. (6) The numbers of waves were adjusted according to the time of the waves and compared among different income levels. (7) Case-number-weighted mortalities among three waves were compared among CRs in different income levels. (8) Mortalities in the largest waves among CRs were compared to investigate the capability of fighting COVID-19 when facing the greatest challenge in CRs at different income levels. (9) Case-number-based normalization was done with the following formula: W(eight) = D * Ci/Ct. Where W = the case-number-weighted death rate, D = death number in a specific wave; Ci = the case number of the same individual CRs, and Ct = the total number of cases in a disease wave. (10) Figures for mortality were visualized using the Chart function in Excel, including those of the Australian Bureau of Statistics, GeoNames, Microsoft, Navinfo, and TomTom.

## 4. Results

### 4.1. Mortality Turning Points in the First Wave of COVID-19 Compared to Other Waves

As of 21 January 2022, more than 100,000 cases were reported from each of the 119 CRs. Among them, only two CRs did not exhibit any distinctive wave (Appendix A). Among the remaining CRs, 55 experienced four waves, 102 showed three waves, 112 had at least two waves and 117 had at least one wave (Figure 1A–D). The average mortalities for the waves 1, 2, 3, and 4 were 0.0359, 0.0194, 0.0168, and 0.0145, respectively. There were no correlations between the first wave and the rest of three waves, with r values of 0.20, 0.06, and −0.04 for waves 2, 3, and 4 (Figure 1E–G). However, there were correlations among the remaining three waves, with r values of 0.57, 0.48, and 0.49 for wave 2 vs. 3, 2 vs. 4, and 3 vs. 4, respectively (Figure 1H–J). The distribution among CRs on mortality rates at the peak points among four waves varies significantly (Figure 1). In the first wave, the high mortality rates were mainly identified in European and North American CRs; conversely, in the other waves, the high mortality rates occurred among developing countries, mainly among African, Southeastern Asian, and South American CRs. 

Appendix A shows the days, and the number of cases and deaths at the turning points of different waves. There are great variations in cases and number of deaths among different countries. It is astonishing to see the extremely high death rate among developed countries. For example, the death rates in France, the UK, Italy, the Netherlands, and Belgium were 21.5%, 18.9%, 14.5%, 13.5%, and 19.7%. These data hinted at a high death rate in the developed countries in the first wave. However, the death rate of the same countries in wave 2 are 1.2%, 1.9%, 2.1%, 0.8%, and 1.2%. These vast changes indicated that better biomedical resources were effectively utilized when the developed countries were ready for the pandemic of COVID-19. In contrast, there is no such a significant change in mortality between wave 1 and wave 2 among developing countries (Figure 1). Therefore, our follow-up analysis focused on the differences among countries at different income levels.

### 4.2. Influential Factors for the Mortalities among CRs

Among 49 potential influencing factors, 9 showed at least one r with an absolute value more than 0.3 (Appendix A, Appendix A). The correlation between the mortality rate and these nine factors showed an opposite direction between the first wave and the subsequent three waves (Figure 2A). In the first wave, factors of categories including economic levels (income, GNI per capita, researchers in R&D, broadband subscriptions, school enrollment), aging population (aged 65 and older and life expectancy), and transportation (owners of passenger cars) are all positively correlated to the mortality rate. These results are consistent with the previous literature findings that economy and transportation boosted the transmission, which caused the high mortalities, particularly within the aged population at the early stage of the COVID-19 pandemic [10,11]. However, in the remaining waves, these factors are either negatively or not correlated to the mortality rate. In particular, the income level, GNI per capita, and researchers in R&D (per million people) (Appendix A) were all negatively correlated to the death rate for waves 2, 3, and 4 (Figure 2B–D). The r values for income for waves 1, 2, 3, and 4 were 0.26, −0.31, −0.26, and −0.33, respectively. The r values for GNI for the waves 1, 2, 3, and 4 were 0.35, −0.29, −0.28, and −0.39, respectively. The values for researchers in R&D for the waves 1, 2, 3, and 4 were 0.37, −0.14, −0.31, and −0.42, respectively. As shown above, these 49 factors are collected from the World Bank, and the data range from 2017 to 2019. Through analysis of all 49 potential factors, we found that only 9 factors show a possible influence on the COVID-19 disease pandemic. Among these nine factors, income level was our focus in this study. Other factors may have their effects and may be analyzed in future studies. For example, passenger cars are closely related to the traveling capacity; as traveling is an important factor for COVID-19 transmission, future studies on this aspect are necessary.

### 4.3. Mortality of COVID-19 at Peak Point Varied across CRs with Different Income Levels

The patterns of mortality rates at the turning points among different income levels were further assessed in the first three waves which contain an adequate number CRs for analysis in groups at different income levels. Compared to the mortalities in wave 1, the reduction rate of the mortalities in waves 2 and 3 were significantly different among CRs at different income levels (Figure 3A). Among CRs at a high-income level, the mortalities in waves 2 and 3 were equal to 29% and 28% of that in wave 1, while in the upper-middle-income CRs, the rate was 76% and 79% for wave 2 and 3. The lower-middle-level CRs had rates of 88% and 90%, while among low-income levels, the rates were 135% and 135% (Figure 3B–E). Thus, the lower the income level, the slower the reduction rate in the mortalities in the later waves of the disease. 

### 4.4. Developing Countries’ Challenges when COVID-19 Reaches the Largest Scale

In order to examine the capacity for fighting COVID-19 when facing the largest challenge, we compared the mortalities among the largest waves in CRs at different income levels. Such a comparison revealed a significant difference between developed and developing CRs (Appendix A). The average mortality in high-income CRs was 0.013, while the average mortalities among upper- and lower-middle-income CRs were 0.021. P values from *t*-tests between high-income CRs and upper- and lower-middle-income CRs were 0.001 and 0.002, while the *p* value between upper- and lower-middle-income CRs was 0.999. The mortality rates among these three categories were different, while the mortality rates within each category were similar (Figure 4A). The mortality rates in the majority of high-income CRs are below 0.02 (Figure 4A), while in the upper- and lower-middle-income CRs, the rates in about 50% of CRs are between 0.02 and 0.04 (Figure 4C,D).

### 4.5. Timeline-Adjusted Comparisons among Different Waves

Due to the variations in the spreading of the disease across the world, the time of the same wave among different CRs differed. Comparison was further conducted on the timeline base for different waves. The waves are re-divided based on the timeline of approximately before July of 2020, between July and November 2020, between November 2020 and March/April 2021, and after, for waves 1, 2, 3, and 4, respectively. The disease patterns among different income levels are compared among these four waves. Data for such a comparison showed similar patterns to the non-time adjusted mortalities among different income levels among the first three waves (Figure 5A). Patterns in different waves indicated that in the high-income CRs, the mortality in the first wave was higher than that in subsequent waves (Figure 5B), while in the CRs with other income levels, the mortalities were higher in the later waves than that in the first wave (Figure 5C,D). 

### 4.6. Comparison of Normalized Data Based on Case Number among Different Waves 

In order to confirm the impact of income levels, we examined the patterns of disease waves using normalized mortality rates based on the case numbers. The normalized data showed that the disease patterns in three waves are similar to that of non-normalized data. The wave 1 mortality rate in high-income CRs was higher than the wave 1 rates in CRs with other income levels, while in waves 2 and 3, the mortality rates in CRs with other income levels were higher than that of the high-income CRs (Figure 5E). Comparing the mortalities of three waves in the CRs indicated that in the high-income level, the rate of the first wave was higher than that of the other waves. On the contrary, the mortality rates in CRs with middle- and low-income levels increased in waves 2 and 3 (Figure 5F and Figure 6). 

### 4.7. Comparison Based on Timeline and Case Normalized Data among Different Waves 

When data were normalized based on both the timeline and case numbers, the patterns of three waves of the pandemics were similar to the non-modified original data. The mortality rate in the first wave in the high-income CRs was much higher than that of the other income categories, while in the third wave, its mortality became the lowest among all income levels (Figure 5I). The distributions of mortality rates of CRs among waves 1, 2, and 3 showed the same patterns (Figure 5J–L).

### 4.8. Wave 4 Preliminary Data and Increasing Lags Based on Income

In our initial evaluation among different income levels, we omitted wave 4 data because there was a significantly smaller number of CRs in the fourth wave compared to the other waves. Instead, we included 19 additional CRs with more than 50,000 cases for a preliminary analysis for wave 4. The majority of these additional CRs are from the African region (Appendix A). Among these 19 CRs, 11 had a visible wave 4 based on our team’s judgment. By adding data from these 11 CRs, we were able to compare the mortality rates among all CRs at four levels of income. Results suggest that the mortality rate in high-income CRs was less than 1% (Figure 6A), while in the upper- and lower-middle-income CRs, the mortalities were as high as 2%. The increased mortality rates in CRs of upper- and lower-middle-incomes reached 100% (Figure 6B). In the low-income CRs, the mortality was higher than 1% with a small number of cases (Figure 6B), which may increase in the future. The distribution of the mortality rates among CRs of these four income levels showed a difference between the CRs of high-income and CRs at the other income levels (Figure 6C–F). 

## 5. Discussion

Our data indicate that conclusions from previous considerable research on the factors influencing COVID-19 rates and deaths were not applicable to the overall COVID-19 pandemic because these results are based on data from the first wave. The occurrence of the first wave has its special characteristics because COVID-19 originated from a metropolis and spread at an unprecedented fast speed through travelling and close contact. Thus, the economic powered convenience of traveling and population density were the major factors [8,9,10] for the disease spreading. It first reached the developed countries and caused high mortality under an urgent and unprepared situation. In comparison, the first wave of the developing CRs happened later than the high-income CRs. It would have caused many more mortalities if the early phase of the pandemic had reached low-income CRs. 

One of the most important findings from this study is that economic disparities affect the capabilities of low-income CRs in fighting the COVID-19 pandemic [10]. The data from waves 2 and 3 confirmed that, in the same situation, developing CRs suffer more than developed countries. There are significant differences in the mortality among different waves of the pandemic in CRs at different income levels. Among high-income CRs, the mortality rates ranged from as high as 5% in wave 1 to approximately 1.4% in wave 3, a 3-fold decrease. Conversely, in the upper- and lower-middle-income CRs, the mortality rates did not decrease during waves 2 and 3. The mortality rates of the CRs with upper- and lower-middle-incomes were approximately 180% of that in high-income countries. Preliminary data from wave 4 support the findings from the previous three waves on the influence of economic disparities in fighting COVID-19. 

The influence of income inequalities on the COVID-19 pandemic in waves 2 and 3 was confirmed by multiple types of analyses due to the complexity of the COVID-19 pandemics across different regions of the world. The data were first shown based on the data from waves of original sequential numbers. This analysis was entirely based on the numbers of waves of the pandemic when comparing among CRs at different income levels. The second comparison was conducted based on the time-sequence of the waves. The third comparison was based on the adjusted data by the number of cases of the peak in the waves. Finally, the data were compared with both timeline- and case-number-adjusted data. All analyses showed the same pattern of negative effects on mortalities by the income levels during waves 2 and 3. Even in wave 4, when the omicron was less severe in symptoms and the world was better prepared with a variety of approaches, the mortalities in the high-income CRs were below 1% and the CRs of upper- and lower-middle-incomes were still as high as 2%. Although data from low-income CRs were missing and incomplete, the mortality was still higher than that of CRs of high-income levels. 

One important comparison in our study is the mortality levels when the pandemic reached the highest case numbers among different waves in CRs of different income levels. Case numbers reaching the maximum number was most challenging to CRs’ capability to fight the COVID-19 pandemic. This comparison clearly demonstrated that the high-income CRs have advantages in the prevention of deaths among their infected population. Different waves happened at different time points. It is important to analyze these different waves at different time points. For example, wave 1 in most of the world occurred before the middle of 2020, while wave 2 occurred during the period between the latter half of 2020 and early 2021, and wave 3 occurred between early 2021 to the middle-to-late of 2021. The importance of time point is not only for the time sequential of the different waves, but also for reflecting the stages of disease variations and virus mutations. The same wave represents the same disease nature and pandemic pattern at a defined period of time. We used average mortality data from each country and conducted a two samples *t*-test to compare the mean of different mortalities. In this way, we could tell the variations of mortality over time.

Two distinguishing features of this study are the analysis of data by waves and the focus on the peaks of the waves. As our data has shown, different waves of the COVID-19 pandemic occurred at different times, and in different regions, environments, and situations. Examining different waves individually reveals various features of these waves. Examining the mortality rate during the peaks of the waves allows us to obtain the data showing the real capability of CRs in fighting the pandemic. This analysis enables us to examine the mortality at the same point from different CRs. In this case, the collected data are comparable.

The positive association between mortality and three other factors in wave 1, ownership of passenger cars (units per thousand persons), percent of population aged 65 and older, and life expectancy at birth, and the non- or negative correlations in other waves of the pandemics provide support that the influential factors for wave 1 do not have the same effect for the rest of the COVID-19 pandemic. In fact, the negative correlations between the car owners and mortality rates in other waves suggest that access to hospitals enhances the chance of survival of patients in the high-income CRs. The consistency of negative correlations between the mortality rates and economic factors such as income level, GNI per capita, and car ownership, indicate that economic levels play essential roles in CRs’ capability to fight COVID-19. 

Due to the limited availability of data, our initial analyses did not include the low-income countries as a separate category when analyzing the influence of income levels in the first three waves. However, based on our preliminary data from seven low-income CRs and a total of more than 12,000 cases, the mortality rate in the low-income CRs was higher than that of the high-income CRs. In considering the data from waves 2 and 3, it is anticipated that the situation in the low-income CRs will worsen when the pandemic reaches the same scale as that of high-income CRs. The longer the pandemic period is, the worse the mortality rates will be in the low-income countries. 

Our study has some limitations. The days of turning point and the waves were arbitrarily determined by our authors, mainly based on the data provided by the Worldometers. The data from Worldometers are not always consistent with that from WHO websites. There were subtle differences in numbers, although these differences did not affect the results. We decided to use the data from Worldometers because of the data convenience as it provides daily numbers of cases and deaths with figures and seven-day averages. It is possible that the data from low-income CRs may not be as complete as that of CRs at other income levels, most likely due to the problems in disease surveillance or data collection. 

Many factors have been reported as influential factors on the mortality of COVID-19. However, our analysis indicated that only a few factors affect mortality. Surprisingly, cases per million and population density did not show an overall significant impact on the mortality rate. The most likely reason is that influential factors in one or more places or countries may not be included in our analysis of 119 CRs. 

As the vaccination and medical treatment develop, the economic advantage of high-income CRs will become more evident. If medical resources cannot be supplied in low- or lower-middle-income CRs, the mortalities in these CRs will increase before they begin decreasing. Because different waves of COVID-19 have different characterizations, the relation between income level and the latest waves of the COVID-19 pandemic may be different from these four waves in our analysis. In particular, omicron has become the dominant virus variant over the world. How the income level affects the pandemic of omicron will be an important question to ask. Furthermore, new variants with new infection characterization and disease features may appear in the future. Therefore, it is essential to monitor the dynamic changes of pandemic pattern with updated information for future studies.

## 6. Conclusions

Our analysis indicated that there are significant differences in disease mortality rates and influential factors between wave 1 and the subsequent waves up to January 2022. Economic disadvantages of developing CRs contributed to more suffering from the COVID-19 pandemic. As time goes on, the vaccination coverage and medical treatment in developed CRs has enabled low mortality rates. The prolonged COVID-19 pandemic causes an increase of mortalities in the developing CRs due to the lack of fighting resources, which are economically dependent.

## 7. Research in Context

**Evidence Before This Study**. The majority reported influential factors on the pandemic COVID-19 are based on the data from the early stage, i.e., the first wave of the pandemic. The few studies on individual waves did not conduct comparison of either multiple waves of the disease or on the disease turning points of the different waves worldwide. 

**Added Value of this Study**. In this study, analysis based on the data from turning points and different waves of the COVID-19 enabled a detailed examination of the relationship between the influential factors and mortalities across CRs around the world. The use of data from the largest wave and the use of time- and case-adjusted data strengthened the finding that economic inequality affects the capacity of lower income CRs to fight the COVID-19 pandemic. 

**Implications of the Available Evidence**. This comprehensive analysis of the mortalities in multiple waves of the COVID-19 pandemic in CRs at different income levels provides a deeper understanding about the influential factors among different waves and the significant role of economic levels in fighting the pandemic. The results suggest that developing CRs may continue to suffer more, while the developed CRs have greatly reduced the death rate from COVID-19 with economic driven research and development on vaccination and treatment. Our data also serves as a warning that serious outcomes in low-income countries may occur in the absence of intervention from the developed CRs. 

## Figures and Tables

**Figure 1 tropicalmed-07-00241-f001:**
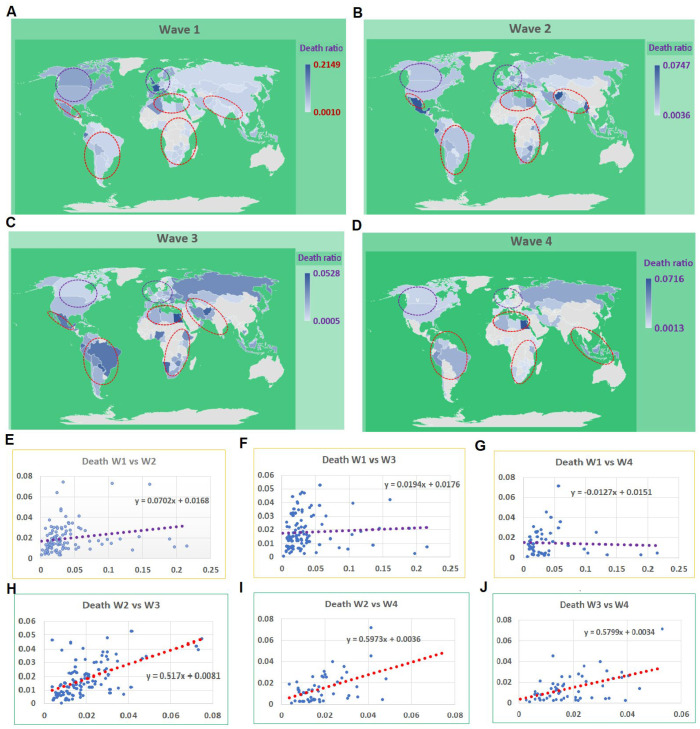
Mortality at the case turning point of the COVID-19 pandemic for 119 countries and regions (CRs). Blue circles indicate the developed CRs, and red circles indicate developing CRs. (**A**–**D**) The mortalities in CRs for waves 1, 2, 3, and 4. In wave 1, the mortalities in developed CRs were higher than other CRs while the mortalities in the waves 2, 3, and 4 in the developing CRs were higher than that in developed CRs. (**E**−**G**) The correlation of mortality between wave 1 and the other three waves, 2, 3, and 4. (**H**−**J**) The correlation among waves 2, 3, and 4.

**Figure 2 tropicalmed-07-00241-f002:**
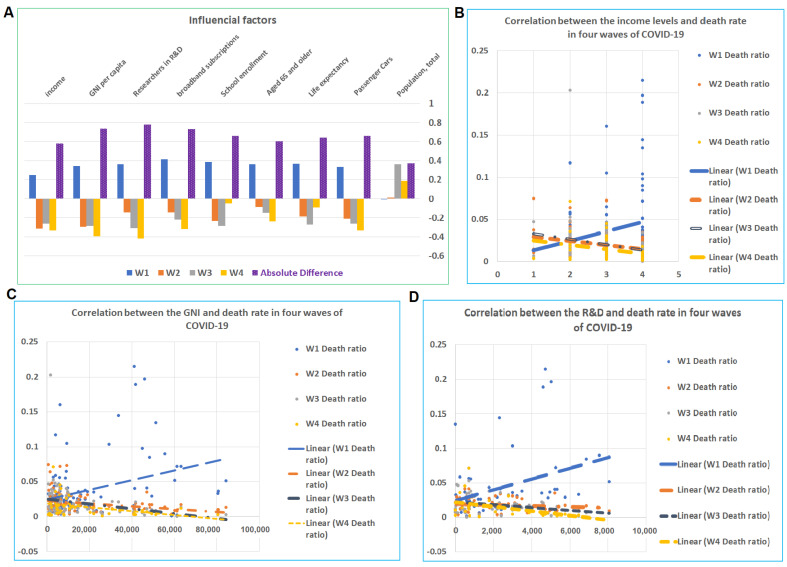
Influential factors on the mortality of COVID-19 during four waves of pandemics. (**A**) Total of nine factors showed r values of more than 3 in at least one wave. The correlations between these nine factors and mortality in wave 1 are in an opposite direction when comparing these correlations with that of waves 2, 3, and 4. (**B**–**D**) The correlations between the mortality in the four waves and income levels (2B), GNI (2C), and R&D (2D). In each case, wave 1 showed positive correlations, while the other waves showed none or negative correlations.

**Figure 3 tropicalmed-07-00241-f003:**
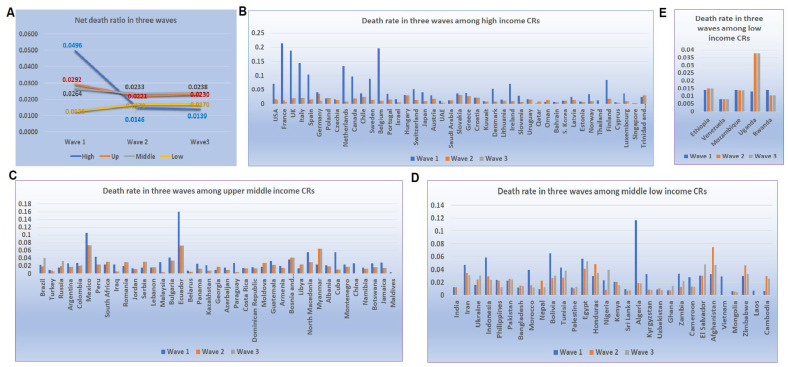
Turning point mortalities during the first three waves of COVID-19 disease in countries according to income levels. (**A**) The average mortalities for waves 1, 2, and 3 in countries of high-, upper-middle-, and lower-middle-incomes, respectively. (**B**) The mortality in three waves in CRs of high-income levels. (**C**) The mortality in three waves in CRs of upper-middle-income levels. (**D**) The mortality in three waves in CRs of lower-middle-income levels. (**E**) The mortality in three waves in CRs of low-income levels.

**Figure 4 tropicalmed-07-00241-f004:**
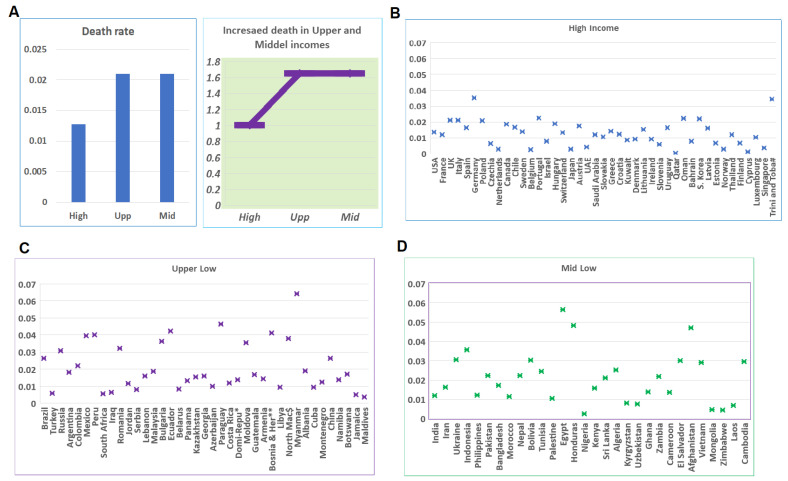
Mortalities in CRs of different income levels at the largest scale of the pandemic. (**A**) Left panel shows the mortality rate in CRs according to income level when COVID-19 reaches the largest scale. Right panel shows the percentages of upper- and lower-middle-income levels in comparison with CRs at a high-income level. (**B**) Distributions of mortalities of CRs at the high-income level. (**C**) Distributions of mortalities of CRs at the upper-middle-income level. (**D**) Distributions of mortalities of CRs at the lower-middle-income level.

**Figure 5 tropicalmed-07-00241-f005:**
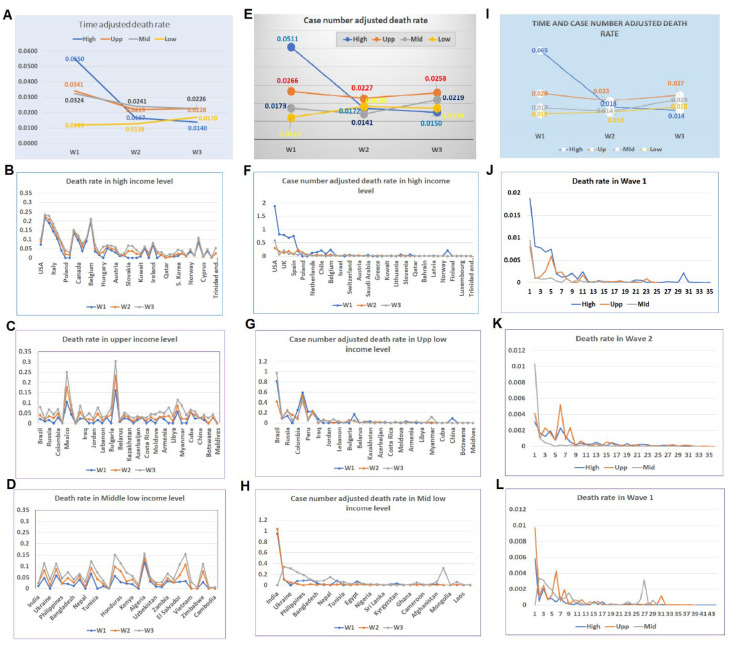
Mortalities based on timeline and case-number-adjusted data. (**A**–**D**) Mortalities based on timeline-adjusted data. (**A**) The average number of deaths in CRs at different income levels in three waves of disease based on the timeline. (**B**) Distributions of mortalities in high-income level CRs by timeline. (**C**) Distributions of mortalities in upper-middle-income level CRs by timeline. (**D**) Distributions of mortalities in lower-middle-income level CRs by timeline. (**E**–**H**) Mortalities based on case-number-adjusted data. (**E**) The average number of deaths in CRs at different income levels in three waves of disease by cases. (**F**) Distributions of mortalities in high-income level CRs by cases. (**G**) Distributions of mortalities in upper-middle-income level CRs by cases. **(H**) Distributions of mortalities in lower-middle-income level CRs by cases. (**I**–**L**) Mortalities based on timeline- and case-number-adjusted data. (**I**) The average number of deaths in CRs at a different income level in three waves of disease by timeline and cases. (**J**) Distributions of mortalities in high-income level CRs by timeline and cases. (**K**) Distributions of mortalities in upper-middle-income level CRs by timeline and cases. (**L**) Distributions of mortalities in lower-middle-income level CRs by timeline and cases.

**Figure 6 tropicalmed-07-00241-f006:**
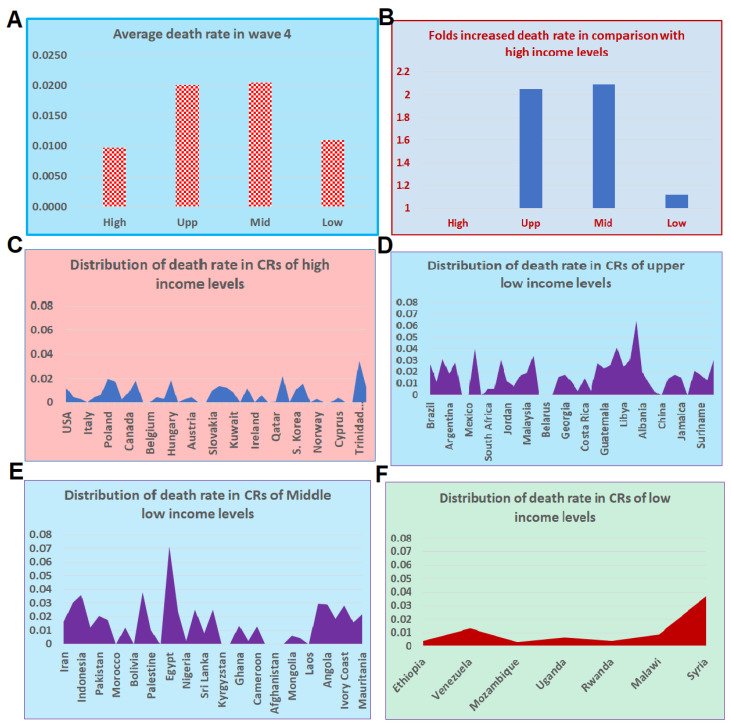
Mortality rates at the turning points during wave 4 of COVID-19 disease in CRs with different income levels. (**A**) Average mortality rates in CRs by income level. (**B**) Fold increase in the upper- and lower-middle, and low income CRs compared to high-income CRs. (**C**) Distributions of mortality rates of CRs at the high-income level. (**D**) Distributions of mortality rates of CRs at the upper-middle-income level. (**E**) Distributions of mortality rate of CRs at the lower-middle-income level. (**F**) Distributions of mortality rate of CRs at the low-income level.

## Data Availability

All COVID-19 cases, deaths, and dates collected are either presented in the article and supplymentary materials or available online at https://www.worldometers.info/coronavirus (accessed on 14 January 2022).

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
