# Peer review of "Mortality in Four Waves of COVID-19 Is Differently Associated with Healthcare Capacities Affected by Economic Disparities"

_tropicalmed, 2022, doi:10.3390/tropicalmed7090241_

Round 1

Reviewer 1 Report

1. In introduction, authors need to highlight more about the importance of doing the analysis. Why did economic disparity matters? In what level: country, households? That would bring the analyses more contextual, and not to mix up between country income level and household income level. What is the most important and authors want to highlight?

2. There is mkixed up between introduction and methods. For example, "We conducted correlation analyses between mortality rates at 74 at peak points in deifferent waves and a total of 49 potential influential factors...." This should be in methods instead of introduction.

3.  The "49 potential influential factors" should be explained in methods section. From which data (review, systematic review)? In what level. It would be better to group them in national or household level. 

4. Why did authors apply Student t-tests while the mortality change over time? Time should be considered in statistical analysis.

5. Figure 1 should be improved. Instead of doing R2 analysis for all countries, it should be better to first group the country based on economic status. It is also related to authors' hypothesis: what did they want you see? If authors want to see the differences between countries' economic level: group them by economic level, and do the comparison. We can find whether the country economic level differed the mortality outcome through regression analyses. The current Figure 1 is too rough and we couldn't conclude anything. Also, the hypothesis should be shaped in the introduction. It is not the matter of the richness of data we have, but what we can learn from the data. 

6. Again, the 49 influential factors are more confusing rather than enlightening. What do we want to learn, with Figure 2. In addition to shaping the hypothesis, authors need to select the most appropriate influential factors that are relevant to the hypothesis. 49 is many, and that are good for data analysis. However, including all of them to analyses could lose us in the middle of nowhere. 

Author Response

Dear Reviewer,

Thank you so much for your constructive comments. We had carefully read your suggestions and did our best to revise the manuscript. The manuscript was improved a lot after major revision. 

Thanks again. 

All authors

Reviewer 2 Report

All comments are in the  file

Author Response

Dear Reviewer, 

Thank you for the constructive comments. We carefully read your suggestions and made revisions as you suggested. 

Thanks a lot.

All authors 

Reviewer 3 Report

The studied theme is very interesting because focuses on mortality in four waves of covid-19 that is differently associated with healthcare capacities affected by economic disparities.

Major comments

1.Research hypotheses were clearly stated, and the results were correlated with them.

2.The methodology should be improved and more accurately described regarding the econometric model used in the paper.

3.The language is very clear and readable.

4.Literature review is adequate to the subject. The novelty of research is clearly presented.

5.I only suggest future research directions may also be highlighted.

Minor comment

1.I suggest ensuring English one more time.

Please verify if is correct the title “Mortality in Four Waves of COVID-19 [are] Differently Associated with Healthcare Capacities Affected by Economic Disparities“, or should be “Mortality in Four Waves of COVID-19 [is] Differently Associated with Healthcare Capacities Affected by Economic Disparities“, taking into account the fact that the subject (‘mortality’) is in the singular, so it is correct that the predicate/verb should also be in the third person singular  (‘is’), resulting ‘mortality is’.

2.The manuscript should respect more accurate the Journal template.

Author Response

Dear Reviewer, 

Thank you for your constructive comments. We revised the manuscript according to your suggestions. 

Thank you so much!

All authors

Round 2

Reviewer 1 Report

No other comments. Authors have improved the paper significantly from my first comments.